# Selenium Status and Hemolysis in Sickle Cell Disease Patients

**DOI:** 10.3390/nu11092211

**Published:** 2019-09-13

**Authors:** Emília Delesderrier, Cláudia S. Cople-Rodrigues, Juliana Omena, Marcos Kneip Fleury, Flávia Barbosa Brito, Adriana Costa Bacelo, Josely Correa Koury, Marta Citelli

**Affiliations:** 1Instituto de Nutrição, Universidade do Estado do Rio de Janeiro, Rio de Janeiro 20550-900, Brazil; emiliadeles@gmail.com (E.D.); claudiacople@gmail.com (C.S.C.-R.); omenaju@gmail.com (J.O.); barbosaflavia@bol.com.br (F.B.B.); jckoury@gmail.com (J.C.K.); 2Faculdade de Farmácia, Universidade Federal do Rio de Janeiro, Rio de Janeiro 21941-590, Brazil; marcos.fleury@yahoo.com.br; 3Instituto Nacional de Infectologia Evandro Chagas, Fundação Oswaldo Cruz 21040-360, Brazil; adribacelo@gmail.com

**Keywords:** selenium, sickle cell disease, human, hemolysis, antioxidant micronutrients

## Abstract

Sickle cell disease (SCD) is a genetic hemoglobinopathy characterized by chronic hemolysis. Chronic hemolysis is promoted by increased oxidative stress. Our hypothesis was that some antioxidant micronutrients (retinol, tocopherol, selenium, and zinc) would be determinant factors of the degree of hemolysis in SCD patients. We aimed to investigate the nutritional adequacy of these antioxidants and their relationships to hemolysis. The study included 51 adult SCD patients regularly assisted in two reference centers for hematology in the State of Rio de Janeiro, Brazil. Serum concentrations of retinol, alpha-tocopherol, selenium, and zinc were determined by high-performance liquid chromatography or atomic absorption spectrometry. Hematological parameters (complete blood count, reticulocyte count, hemoglobin, direct and indirect bilirubin, total bilirubin, lactate dehydrogenase) and inflammation markers (leukocytes and ultra-sensitive C-reactive protein) were analyzed. A linear regression model was used to test the associations between the variables. Most patients presented selenium deficiency and low selenium consumption. Linear regression analysis showed that selenium is the main determinant of hemolysis among the antioxidant nutrients analyzed. Thus, data from this study suggest that the nutritional care protocols for patients with SCD should include dietary sources of selenium in order to reduce the risk of hemolysis.

## 1. Introduction

Sickle cell disease (SCD) is a hereditary disorder caused by a mutation in the gene encoding the β-hemoglobin subunit (β-Hb), which gives rise to hemoglobin S (HbS) [1]. Under low oxygen concentrations, HbS loses oxygen (deoxy-HbS), changes its globular structure, and forms polymers. This also leads to alterations in the discoid morphology of erythrocytes, which assume a sickled shape and become more rigid and more susceptible to hemolysis [2]. HbSS and HbSC are the most common genotypes in SCD. HbC is another single mutation/amino acid change at the same position in β-Hb, but with lysine replacing glutamic acid. Therefore, the HbSC genotype derives from the simultaneous occurrence of mutations (Glu6 → Val; HbS) and (Glu6 → Lys; HbC) in each of the alleles of the gene encoding β-Hb. The homozygous HbSS genotype is associated with the most severe form of SCD, named sickle cell anemia (SCA).

Chronic hemolysis is one of the pathophysiological features of SCD, which reduces erythrocyte lifespan by approximately 70% [3]. This process is related to increased oxidative stress in SCD [4]. HbS instability and auto-oxidation contribute to the overproduction of intracellular reactive oxygen species (ROS), such as the hydroxyl radical (OH·), generated by the Fenton reaction [5,6]. In addition, the release of heme, hemoglobin, and iron in the plasma due to hemolysis triggers a ROS increase in the extracellular environment. The presence of iron and heme in the plasma potentiates the Fenton reaction, damaging proteins and polyunsaturated lipids and causing instability of the erythrocyte membrane [7]. This process is cyclical and depletes the antioxidant systems, ultimately leading to oxidative stress [4].

Some nutrients such as vitamins A, C, E, selenium, and zinc play an important role as part of the antioxidant defense system in minimizing the deleterious effects of ROS [8]. Vitamin A acts by inhibiting the spread of lipid peroxidation, and low serum vitamin A was shown to be associated with increased oxidative stress products and smaller concentrations of reduced glutathione [9]. Vitamin E prevents lipid peroxidation by hindering continuous oxidative chain reactions in membranes and lipoproteins [10] and regenerates vitamin C, preserving its antioxidant role [8]. Zinc is a cofactor of the superoxide dismutase (SOD) enzyme. It inhibits the NADPH oxidase enzyme, helps stabilize sulfhydryl proteins, and antagonizes redox-active metals, such as copper and iron, which catalyze the formation of free radicals through the Fenton reaction [11]. Selenium is present at the active site of glutathione peroxidase (GPx) [12] as a constituent of the amino acid selenocysteine. It maintains the activity of this antioxidant enzyme, protecting hemoglobin against oxidation inside the erythrocytes, thus making them less susceptible to hemolysis [13]. Additionally, selenium plays an important role during erythropoiesis, in which selenoproteins influence the multiple stages of erythroid development, in addition to controlling the oxidative stress that occurs during this process, thus promoting adequate cell maturation [14].

Considering the abovementioned functions, it is plausible that the adequate nutritional status of these micronutrients may protect erythrocytes from hemolysis, particularly in older adults [15,16], whose antioxidant defense is commonly depleted. Life expectancy of individuals with SCA has increased and, in Brazil, the current estimate is 53.3 years of life [17]. Possibly, for this reason, this population is poorly studied. Conversely, deficiency of these micronutrients may contribute to increased oxidative stress and erythrocyte susceptibility to hemolysis. For this reason, we hypothesized that some antioxidant micronutrients (retinol, tocopherol, selenium, and zinc) would determine the degree of hemolysis in SCD adult patients. Therefore, we aimed to investigate the nutritional adequacy of these antioxidants and their relationships to hemolysis. The results may support the development of a better nutritional management for SCD patients, which may improve their clinical status.

## 2. Materials and Methods

### 2.1. Participants

This study was conducted from 2012 to 2014 and involved 51 adult SCD patients who met the following criteria: age ≥ 40 years (both sexes); presence of the HbSS and HbSC genotypes; absence of chronic use of immunosuppressive drugs, barbiturates, corticosteroids, and thyroid hormone replacement; no diagnosis of liver disease; no use of vitamin and/or mineral supplements during the last 60 days (except for folic acid); no addiction to narcotics and alcohol; absence of any cognitive impairment that could prevent the collection of the requested information. Patients who had been hospitalized and/or those who received blood transfusions 15 days prior to blood collection were not included in this study. 

### 2.2. Blood Collection

Whole blood was collected after 8–10 h of fasting in tubes containing the anticoagulant agent ethylenediaminetetraacetic acid (EDTA) or a clot activator gel, or in mineral-free tubes to analyze trace elements. All biochemical and laboratory analyses were performed at the Clinical Analysis Laboratory of the School of Pharmacy (LACFAR), Federal University of Rio de Janeiro (UFRJ).

### 2.3. Hematological Parameters and Hemolysis Markers

The hematological parameters assessed were complete blood count, reticulocyte count (RET), hemoglobin (fetal hemoglobin (HbF) and HbS) level, and serum concentrations of direct bilirubin (DB), total bilirubin (TB), indirect bilirubin (IB), and lactate dehydrogenase (LDH). The complete blood count was performed using the Horiba Pentra 60 C+ analyzer (HORIBA ABX SAS, ABX Pentra 60, São Paulo, Brazil). Reticulocytes were analyzed using the brilliant cresyl blue staining technique. Serum DB and TB were analyzed using a colorimetric method (Labtest, LabmaxPlenno, Belo Horizonte, MG, Brazil). IB was estimated by calculation, from the subtraction of the value of DB from that of TB. Serum LDH concentration was analyzed using the continuous ultraviolet kinetics method (Labtest, LabmaxPlenno, Belo Horizonte, MG, Brazil). Hemoglobin variants and quantifications were determined using cation-exchange high-performance liquid chromatography (HPLC), by using the VARIANT II Hemoglobin Testing System (VARIANT^®^, Bio-Rad Laboratories, Hercules, CA, USA) [18]. Each hemoglobin variant has its characteristic retention time, established for common variants using the Variant short program.

### 2.4. Inflammatory Markers

Serum ultra-sensitive C-reactive protein (us-CRP) concentration was analyzed by nephelometry (Labtest, LabmaxPlenno, Belo Horizonte, MG, Brazil), and the leukocyte counting was performed using the Horiba Pentra 60 C+ analyzer (HORIBA ABX SAS, ABX Pentra 60, São Paulo, Brazil). 

### 2.5. Antioxidant Micronutrients

HPLC (Agilent Technologies Brazil Ltd.a., Agilent 1290 UHPLC System, São Paulo, Brazil) was used for retinol and α-tocopherol serum concentrations analysis. A silica column measuring 5 mm, 15 cm × 6 mm (Shim-Pack, Shimadzu), was used in the chromatographic phase. The mobile phase consisted of 99:1 hexane/isopropanol, an isocratic elution system, and a flow of 1.8 mL/min. A UV–Vis detector (Waters 996, Eschborn, Germany) was used for peak and spectrum characterization. It was operated at 325 nm (retinol) and 295 nm (α-tocopherol). Standard curves were generated with external standards for retinol and α-tocopherol (Sigma-Aldrich, Darmstadt, Germany). Serum concentrations of selenium and zinc were analyzed using atomic absorption spectrometry (Agilent, AA 240/280 Series Spectrometer, São Paulo, Brazil).

### 2.6. Cut-Off Values to Determine Micronutrient Deficiency

The cut-off values used to determine antioxidant micronutrient deficiency were <0.35 μmol/L for retinol [19], <5 μg/mL for α-tocopherol [20], <80 μg/dL for selenium [21], and <70 μg/dL for zinc [21].

### 2.7. Food Consumption

Adequacy of vitamin A, E, zinc, and selenium intake was determined according to the dietary reference intake (DRI) [22]. Participants provided detailed dietary intake information for two 24 h periods spaced at least 10 days apart. Accurate descriptions and portion sizes of foods consumed were facilitated through the use of food pictures and diagrams. Each participant’s usual intake was obtained through averaging the two 24 h dietary intakes. All 24 h recalls were administered in person. The diet analysis program DietWin^®^ Professional Plus (DietWin, Porto Alegre, Brazil, 2015) was used to determine the intake of food group portions and specific nutrients.

### 2.8. Statistical Analysis

Statistical analyses were performed using the Statistical Package for Social Science software (IBM SPSS^®^ Inc., version 22.0, Chicago, IL, USA). The distribution of continuous variables was analyzed using the Shapiro–Wilk test. Data were described as median, minimum, and maximum values and 25th to 75th percentile, due to the asymmetric distribution of the variables.

Backward linear regression analysis was used to test the significance level in order to identify the relationship between plasma concentrations of antioxidant micronutrients (retinol, α-tocopherol, selenium, and zinc) and hematological parameters (erythrocytes, hemoglobin, reticulocytes, serum LDH, DB, IB, and TB, HbF, and hemoglobin S). Multiple regression analyses were performed individually using a set of dependent variables (hematological parameters) to identify combined associations. Thus, serum antioxidant micronutrients concentrations were considered as independent variables. The control variables were sex, age, ethnicity, genotype, use of hydroxyurea, serum us-CRP concentration, and leukocyte count. Results are presented as B unstandardized coefficients, with the corresponding 95% confidence intervals (95% CI) and *p* values. A significance level of 5% was adopted for all statistical analyses.

### 2.9. Ethical Aspects

All participants were informed about the study’s objectives and provided written informed consent. The study was approved by the ethics committees of the State Institute of Hematology Arthur de Siqueira Cavalcanti (HEMORIO), technical opinion number 244/10, and of the Pedro Ernesto University Hospital (HUPE), technical opinion number 2819/2010. Therefore, the study was performed in accordance with the ethical standards laid down in the 1964 Declaration of Helsinki and its later amendments.

## 3. Results

### 3.1. Participants’ Characteristics 

The characteristics of this study’s participants are shown in Table 1. Of the 51 patients studied, 36 (70.6%) had the HbSS genotype. Mean age was 49.3 years old (±6.57), and most patients were female (54.9%). Regarding skin color, there was a higher prevalence of black patients (52.9%). Most patients (58.8%) did not use the myelosuppressive agent hydroxyurea.

Table 2 shows the descriptive data of hematological parameters, revealing the anemic characteristics of this population.

Table 3 shows the descriptive data of inflammatory parameters and the serum concentrations of antioxidant micronutrients. The results are in accordance with the low-grade chronic inflammation usually observed in SCD. The median concentrations of vitamin A, E, and zinc were above the reference values. On the other hand, selenium median concentration was below the reference value. Accordingly, most patients had adequate serum concentrations of zinc (95.0%), vitamin E (91.8%), and vitamin A (73.5%), while a high rate (93.5%) of inadequate serum concentration of selenium was observed. 

### 3.2. Relationship between Antioxidant Micronutrients and Hematological Parameters

Selenium is directly associated with hemolysis in SCD (Table 4). Positive associations between serum concentration of selenium and erythrocyte counts, hemoglobin level, and hematocrit were observed. Serum selenium concentration was inversely associated with hemolysis markers as shown by the result of reticulocyte count and by serum LDH, TB, DB, IB, and HbF concentrations.

Vitamin E was inversely associated with red cell distribution width (RDW) (*p* = 0.005). The other micronutrients did not present significant association with hematological parameters. There was no significant difference between antioxidant micronutrients concentration when comparing SS and SC genotypes (data not shown).

When redoing the analyses without any adjustment (data not shown), selenium associations with the hemolysis markers (LDH, erythrocytes, hemoglobin, hematocrit, TB, IB, DB, reticulocyte) were maintained, except for HbF (no association of HbF with selenium in the unadjusted model). Moreover, in this unadjusted model, new associations with other nutrients were observed. There was an indirect association between vitamin A and LDH (*p* = 0.013) as well as between zinc and BD (*p* = 0.014), vitamin E and BD (*p* = 0.049), reticulocyte and vitamin A (*p* = 0.021), and HbS and vitamin E (*p* = 0.007).

### 3.3. Food Consumption

Figure 1 shows the adequacy of antioxidant micronutrients consumption. Most patients consumed inadequate quantities of vitamin A (65%), vitamin E (80%), selenium (82%), and zinc (82%). Two percent (2%) of the individuals studied presented consumption above the tolerable upper intake level (UL) of vitamin A.

## 4. Discussion

Oxidative stress influences hemolytic events [1], but the relationship between the levels of antioxidant micronutrients and hemolysis in SCD patients is not well explored. In this study, we identified the relationship of serum concentrations of retinol, alpha-tocopherol, selenium, and zinc with hemolysis markers. 

In our study, most patients had selenium deficiency and adequate nutritional status of the other analyzed antioxidants. Selenium deficiency may possibly be attributed to a selenium-poor diet, since the food sources of selenium are expensive and, in Brazil, the population is more vulnerable to food insecurity [24]. In addition, selenium deficiency may also be attributed to the increased renal excretion of SCD patients. Individuals in the age group studied are more susceptible to the development of chronic renal failure [25], which is related to lower selenium concentrations [26]. In addition, tubular reabsorption is known to be abnormal in individuals with SCD, leading to loss of nutrients due to the repeated sickling process of red blood cells [27]. 

Despite the low consumption of vitamins A and E, most SCD patients were not deficient in these nutrients, perhaps because these are fat-soluble vitamins stored in the body. Thus, 24 h dietary recalls may not directly reflect the nutritional status of these nutrients. However, the median values observed in Table 3 for these vitamins are close to the cutoff point used. The mean vitamin A concentration was 0.5 µmol/L, while the cutoff point was 0.35 µmol/L, which is the lowest cutoff point used by WHO to determine vitamin A deficiency. For vitamin E, the median was 7.9 µg/mL, while the cutoff used was 5 µg/mL. 

Claster et al. [28] evaluated the prevalence of vitamin and mineral deficiency in American SCD patients with iron overload and thalassemia and showed a higher percentage of abnormal low serum vitamin A (73.70%) and selenium (65.50%) concentrations in SCD patients (the authors do not describe which genotypes were analyzed). In that study, different hemoglobinopathies were mixed together. We found no differences in nutrient concentration between the genotypes HbSS and HbSC (data not shown).

Selenium is directly associated with hemolysis in SCD. This micronutrient is part of the structure of selenoproteins, such as GPx, which are present in the erythrocytes and help protect hemoglobin against oxidation [29]. In addition, an adequate selenium intake preserves GPx activity [30]. Thus, we suggest that reduced serum concentrations of selenium may be associated with impaired antioxidant capacity of erythrocytes caused by reduced activity of GPx and other selenoenzymes, as shown by Natta and colleagues [31], who observed low serum selenium concentrations and lower GPx activity in patients with SCD. That study suggests that there is low antioxidant capacity in these individuals, making proteins and lipids of the erythrocyte membrane more susceptible to oxidation and subsequent hemolysis [31]. 

As far as we know, only one study has examined the association between selenium and hemolysis markers [32]. That study examined thalassemia major and SCD children patients and analyzed only LDH and reticulocytes as hemolysis parameters. Although these patients had lower serum selenium concentrations compared to the control group, there was no significative associations between selenium and hemolysis markers. 

In addition, recent studies have demonstrated the effect of selenium on erythropoiesis [14,33]. Selenium deficiency reduces the activity of the transcription factor GATA-1 (GATA-binding protein 1), responsible for erythroid differentiation, and compromises the transport of heme in the erythroblastic islands, damaging the terminal maturation of erythroblasts [14]. This suggests that the role of selenium in erythropoiesis goes beyond that of a protective antioxidant. Animals that received adequate amounts of selenium were able to recover from anemia, whereas maintenance of selenium deficiency was shown to be lethal [14], which is in accordance with our study, where the selenium nutritional status significantly influenced erythrocyte counts, hemoglobin levels, and hematocrit, pointing to a direct association between these variables and serum concentrations of selenium. Thus, in SCD patients, it is possible that selenium simultaneously contributes to increase erythropoietic effectiveness and to decrease hemolysis rates. Our results also raise the possibility that other hemolytic anemias may be benefited by the nutritional adequacy of selenium. As already discussed by other authors, selenium is still poorly understood in terms of its biomedical importance [34]. It is known, however, that the nuclear factor E2-related factor 2 (Nrf2) is a key transcription factor binding to the antioxidant response elements (ARE) present in the promoter regions of various antioxidant enzymes. In selenium deficiency, Nrf2 expression is increased, as a compensatory mechanism [33,35]. In sickle erythroid cells, Nrf2 induces HbF synthesis in order to inhibit hemoglobin S polymerization and protect against oxidative stress due to chronic hemolysis [35]. This may explain the inverse relationship between selenium and HbF, shown in Table 4. At the same time, selenium reduction increases inflammation, which usually reduces HbF concentrations. Therefore, when we removed the effect of inflammation (adjusted model; Table 4), we observed an inverse association between selenium and HbF. However, when the model was not adjusted for inflammation (data not shown), we no longer saw this association. In addition, it is known that in SCD hydroxyurea enhances HbF production [2], which explains the elevated levels of HbF observed in the present study. 

We observed an inverse relationship between vitamin E and RDW, which may be explained by vitamin E activity, which protects the fatty acids of the erythrocyte membrane, reducing the membrane susceptibility to lipid peroxidation [36] and subsequent hemolysis [37].

In our study, most patients had an adequate nutritional zinc status, possibly due to extravasation of erythrocyte zinc (concentration 10 to 20 times greater than the amount in plasma) because of hemolysis observed in SCD patients [38]. This and the absence of a control group are limitations of the present study. Despite the antioxidant action of zinc, no association was observed between serum zinc concentrations and hemolysis. BAO et al. [39] showed that zinc supplementation was effective to reduce oxidative stress in SCD patients and improve erythrocyte count, hemoglobin levels, and hematocrit. The authors suggest that increased red blood cell indexes may be related to the antioxidant effect of zinc, but its influence on GATA-1, a zinc-dependent erythroid transcription factor, should be considered [39,40].

Altogether, our results allow the proposition of the cyclical mechanism presented in Figure 2. We suggest that a reduced serum concentration of selenium may impairs the body’s antioxidant capacity, resulting in higher ROS formation and subsequent oxidative stress. Increased ROS levels may damage the erythrocyte membrane, making these cells more susceptible to hemolysis. Hemolysis products (hemoglobin, heme, iron), in turn, can stimulate leukocyte recruitment and production of proinflammatory cytokines in the vascular endothelium, triggering an inflammatory process. Therefore, inflammatory cells may produce more ROS, worsening the oxidative stress and leading to hemolytic events [41].

## 5. Conclusions

Among the antioxidant micronutrients studied, selenium was found to play the most important role in hemolysis in SCD patients. Low serum concentrations of selenium were directly associated with hemolytic events, which may aggravate the condition of these patients and result in more severe complications.

An adequate selenium intake may probably improve SCD patients’ clinical status. Thus, we recommend the inclusion of an increased amount of selenium-rich foods in their diet, in order to increase the concentration of this micronutrient and reduce pro-oxidant and hemolytic processes in SCD. 

## Figures and Tables

**Figure 1 nutrients-11-02211-f001:**
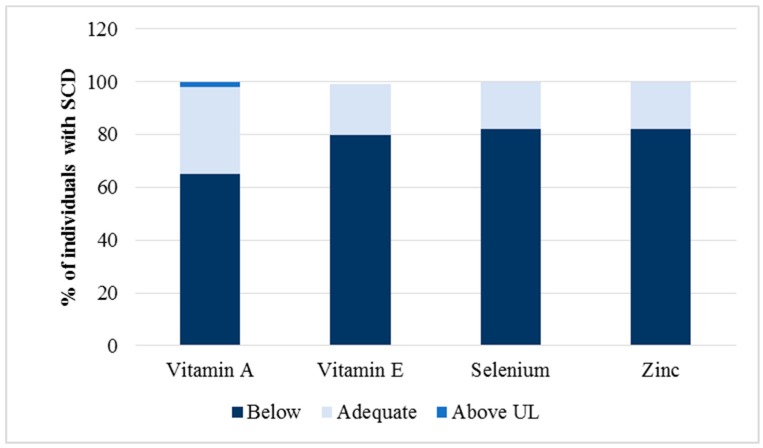
The adequacy of antioxidant micronutrient intake was assessed by 24 h dietary recalls, based on the dietary reference intake (DRI) recommendations [22].

**Figure 2 nutrients-11-02211-f002:**
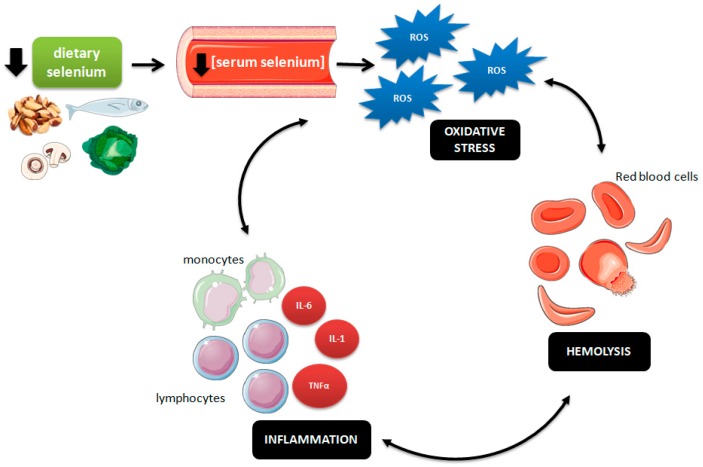
Representative scheme of the relationship between low dietary selenium intake, serum selenium concentration, and the oxidative–hemolysis–inflammation cycle in sickle cell disease (SCD) patients. ROS (reactive oxygen species), IL-1 (interleukin 1), IL-6 (interleukin 6), TNFα (tumor necrosis factor α).

**Table 1 nutrients-11-02211-t001:** Characteristics of adult sickle cell disease patients (*n* = 51).

	*n* (%)
Genotype	
HbSS	36 (70.6%)
HbSC	15 (29.4%)
Sex	
female	28 (54.9%)
male	23 (45.1%)
Age (mean ± SD ^a^)	49.3 ± 6.57
Skin colour	
White	2 (3.9%)
Black	27 (52.9%)
Mixed race	22 (43.1%)
Use of hydroxyurea	
Yes	21 (41.2%)
No	30 (58.8%)

HbSS (hemoglobin S homozygous genotype of sickle cell disease, called sickle cell anemia), HbSC (hemoglobin S and hemoglobin C heterozygous genotype of sickle cell disease, called HbSC disease). ^a^ Standard deviation.

**Table 2 nutrients-11-02211-t002:** Hematological parameters in adult sickle cell disease patients (*n* = 51).

	Reference Value *	Median	Minimum; Maxi mum	P25; P75
ERY (10^12^/L)	Fem 4.0–5.2	2.83	1.63; 5.03	2.31; 3.73
	Male 4.5–5.9			
Hb (g/dL)	Fem 12–16	9.10	5.50; 14.70	7.50; 10.10
	Male 13.5–17.5			
HT (%)	Fem 36–46	27.8	16.70; 46.40	23.40; 32.10
	Male 41–53			
MCV (fl)	80–100	96.00	63.00; 135.00	86.00; 102.00
MCH (pg)	27–32	31.00	19.00; 44.10	27.30; 33.20
RET (% red cells)	0.5–2.5	5.50	1.60; 18.30	0.30; 9.40
RDW (%)	11.5–14.5	15.00	11.10; 20.90	13.20; 17.70
TB (mg/dL)	0.3–1.0	1.37	0.50; 7.28	0.94; 2.51
DB (mg/dL)	0.1–0.3	0.58	0.20; 1.91	0.32; 0.86
IB (mg/dL)	<0.8	0.96	0.10; 5.97	0.62; 1.70
LDH (U/L)	100–190	818.50	119.00; 2594.00	541.00; 1127.00
HbF (%)	0–2.0	4.40	0.20; 27.40	1.20; 12.40
HbS (%)	–	77.35	23.50; 95.10	49.40; 88.42

ERY (erythrocytes), Hb (hemoglobin), HT (hematocrit), MCV (mean corpuscular volume), MCH (mean corpuscular hemoglobin), RET (reticulocytes), RDW (red cell distribution width), TB (total bilirubin), DB (direct bilirubin), IB (indirect bilirrubin), LDH (lactate dehydrogenase), HbF (fetal hemoglobin), HbS (hemoglobin S). * See reference [23].

**Table 3 nutrients-11-02211-t003:** Serum concentrations of antioxidant nutrients and inflammation markers in sickle cell disease patients (*n* = 51).

	Reference Value *	Median	Minimum; Maximum	P25; P75
Vitamin A (µmol/L)	>0.35	0.50	0.20; 1.30	0.30; 0.80
Vitamin E (µg/mL)	>5	7.90	4.20; 20.6	6.70; 11.45
Selenium (µg/dL)	>80	33.50	15.00; 130.00	27.00; 41.00
Zinc (µg/dL)	>70	96.00	62.80; 145.70	83.70; 117.20
Leukocytes (cells/mm³)	4500–10,000	8200	3500; 25900	6600; 10200
us-CRP (mg/dL)	<0.5	0.55	0.09; 2.79	0.25; 1.13

us-CRP (ultra-sensitive C-reactive protein). * See references [19,20,21,23].

**Table 4 nutrients-11-02211-t004:** Final model between nutrients and hematological parameters in adult sickle cell disease patients (*n* = 51).

Antioxidant Nutrients	Hematological Parameters	*R* ^2^	*β*	CI	*p* Value ^a^
Selenium	LDH (U/L)	0.62	−13.90	−22.79; −5.02	0.003
	ERY (10^12^/L)	0.30	0.02	0.00; 0.04	0.010
	Hb (g/dL)	0.38	0.06	0.01; 0.10	0.006
	HT (%)	0.37	0.19	0.06; 0.32	0.006
	RET (%)	0.27	−0.09	−0.19; 0.00	0.040
	TB (mg/dL)	0.54	−0.05	−0.08; −0.02	0.001
	DB (mg/dL)	0.56	−0.01	−0.02; 0.00	0.007
	IB (mg/dL)	0.42	−0.04	−0.07; −0.01	0.007
	HbF (%)	0.41	−0.22	−0.40; −0.04	0.017
Vitamin E	RDW (%)	0.28	−0.26	−0.43; −0.08	0.005

CI (confidence interval). Only significant results from the backward linear regression analysis are shown in this table. ^a^
*p* values obtained from the backward linear regression analysis, considering antioxidant micronutrients (vitamin A, E, selenium, and zinc) as independent variables and hematological parameters (LDH, ERY, Hb, HT, RDW, RET, TB, DB, IB, HbF, and HbS) as dependent variables. The analysis was adjusted for the following variables: skin color, age, sex, genotype, use of hydroxyurea, us-CRP, and leukocyte count.

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
