# Peer review of "Selenium Status and Hemolysis in Sickle Cell Disease Patients"

_nutrients, 2019, doi:10.3390/nu11092211_

Round 1

Reviewer 1 Report

This an interesting paper, I would like to see a follow up paper to determine if there is clinical improvement in the participants following dietary increase in selenium.

Introduction 

First paragraph requires review it is written in a confusing and disjointed way and does not provide a good summary of sickle cell disease (SCDSCD). I think the focus is on Hb SS and Hb SC because these are the participants investigated in this study, but this is not clear. There are other genotypes which result in SCD some of which give an appearance similar to Hb SS using the HPLC technique described in the methods. 

Line 42, I don't believe it is fair to say the the most important pathophysiological characteristic of SCD is chronic haemolysis. It is generally accepted there are three main physiological processes, polymerisation, VOC and haemolysis. 

Final paragraph of the introduction also requires some clarification, it is difficult to follow what the authors message is. What is meant by the term "elder adults"? 

Materials and Methods

Has the patients genotype been confirmed by molecular analysis. Are the participants on transfusion or exchange transfusion programmes? 

There is no control group which is a study limitation.

Results

I note Hb F levels varied in participants from 0.2 - 27.4, what is the reason for the higher levels, HU therapy or HPFH? Also the Hb S levels vary from 0 to 95%. Please explain a level of 0% for Hb S. Is there any relationship between the variation in the levels and the micronutrient levels. Effect of transfusions if given has not been discussed. 

Line 178. It must be clarified that this study shows that selenium is most directly associated with haemolysis, it cannot be said it is the main determinant. This sentence is repeated in line 211. Both need to be corrected.

Conclusion

Lines 237 - 243, the authors state that the Claster et al study mixed different genotypes and hemoglobinopathies, Claster studied thalassaemia major and SCD. The SCD genotypes were not clarified. The current study has confirmed that Hb SC has been distinguished from Hb SS but has not clarified the genotypes of the Hb SS patients. This needs to clarified here.

Please ensure that consistency of terminology is used the paper mostly writes selenium in full but on line 229 it is written as Se.

Reviewer 2 Report

This is an interesting piece of work that will be of interest to the readers of Nutrients. I have a couple of questions and suggestions that might improve the paper.

Major:

- The authors find an intriguing, apparently selective association of selenium with different hemolysis parameters in Table 4. In this analysis, a total number of seven parameters were adjusted for (“skin colour, age, sex, genotype, use of hydroxyurea, us-CRP, and leukocyte count”).

First, how important, in quantitative terms, were these adjustments? What about the unadjusted data? Did the selenium association occur?

Second, I understand the adjustment for age, sex, etc. However, I find it difficult to see the point of adjusting for CRP and leukocyte count, as those are causally (!) influenced by hemolysis and thus by selenium status, as the authors argue. Hence, we would have a classic case of “circular reasoning” here. I urge the authors to present an additional table in which CRP and leukocyte count are not adjusted for, in addition to some concise information about the entirely unadjusted, raw data.

- There are a number of articles in Pubmed reporting on selenium in SCD, none of which is cited and discussed by the authors I believe. Please check those articles and give reference to the relevant ones. This is a shortage of the current manuscript.

Minor:

- Why were so many (80%) of the patients nutritionally deficient on zinc, selenium, etc. (Figure 1)? This is an unusually high number that requires an explanation.

- Do the authors have a hypothesis why only selenium was mostly deficient in serum (Table 3) while the patients’ nutritional status was about equally deficient for selenium, zinc and vitamin E? Selenium is not consumed faster under conditions of oxidative stress, whereas vitamin E is indeed consumed faster (through chemical degradation). Hence, I would have expected the opposite result in SCD.

- I find Figure 2 not very helpful. It rather confuses what the authors want to say. In my words, “selenium deficiency (of unknown cause) appears to exacerbate hemolysis and subsequently inflammation in genetically caused SCD”. Hence, the double-tipped, pseudo-circular arrows are obfuscating things. Please redraw, considering that “vicious circles” are not a scientific concept.  

- Please amend the following phrases:

Line 59, “Selenium is the main constituent of the glutathione peroxidase”. This is not correct.

Line 232, “due to the vitamin E activity, which restores intracellular cations”. Vitamin E does not interact with cations.

Line 246, “This is in contrast with the results of BAO et al.”. Zinc is a minor antioxidant at best; most people in the field would not classify it as such. Hence, I do not see a contrast here.   

- Finally: Do the authors have patient blood samples stored to potentially measure GPx activity? Such measurements would have been very revealing.
